# Grouting mechanism and experimental study of goaf considering filtration effect

Lei Zhu[1], Wenzhe Gu[1,2]*, Yibo Ouyang[3], Fengqi Qiu[1]

**1** China Coal Energy Research Institute Co., Ltd., Xi'an, China, **2** School of Energy and Mining Engineering, China University of Mining and Technology(Beijing), Beijing, China, **3** School of Energy, Xi'an University of Science and Technology, Xi'an, China

* 274754782@qq.com

**Data Availability Statement:** All relevant data are within the paper.

**Funding:** The author(s) received no specific funding for this work.

## Abstract

The filtration effect significantly affects the gangue slurry velocity and concentration, making it difficult to evaluate the gangue slurry diffusion range. Based on the Darcy seepage law, a seepage theoretical calculation model is established considering the filtration time and space effect. And the "water-cement ratio change matrix" in the seepage process of coal gangue slurry is deduced, revealing the basic mechanism of the porous media filtration effect, and the water-cement ratio gradually increases in the seepage process of gangue slurry. The visual test platform for slurry diffusion in goaf was independently developed for testing. The active heating optical fiber method (AHFO) was used to monitor the flow and diffusion of coal gangue slurry in the collapse zone of goaf, and the gravity gradient and water cement ratio of slurry in goaf were measured. The law of particle sedimentation in the gangue slurry flow process under the filtration effect was revealed, and engineering verification was carried out. The results show that the average slope of the gangue slurry in the gangue accumulation is 6.34%, and the overall flow law of the gangue slurry in the goaf is the first longitudinal expansion and then transverse diffusion. The water-cement ratio near the grouting mouth is smaller than the initial water-cement ratio, the near-end water-cement ratio is smaller, and the far-end water-cement ratio is larger. During on-site filling, the accumulated grouting volume of a single hole is 700 m³, and the gangue slurry diffusion distance is greater than 45m, indicating that the gangue slurry has good fluidity.

## 1 Introduction

Large-scale and high-intensity mining activities and the utilization of coal resources have produced a large amount of solid waste of coal gangue [1–3], occupying a large number of land resources and polluting air, groundwater and soil, resulting in a series of ecological, environmental problems [4–6]. The coal gangue and water are mixed into a certain concentration of gangue slurry through the underground slurry preparation system, and the goaf caving zone is filled with adjacent grouting to realize the harmless disposal of coal gangue under the low interference condition of the working face [7, 8]. The filtration effect mainly occurs in the flow process of suspension in porous media. When the slurry is injected into the rock and soil

**Competing interests:** The authors have declared that no competing interests exist.

mass, some cement particles will be blocked by the particle skeleton, which will gradually filter out the cement particles, leading to the decrease of grout concentration, the blockage of pores, and the increase of grouting difficulty, which is called filtering effect [9–11]. The filtration effect generally exists in the seepage grouting of granular rock and soil, which plays an important role in seepage grouting [12–14]. The filtration effect significantly affects the gangue slurry velocity and concentration, making it difficult to evaluate the gangue slurry diffusion range.

The filtration effect in the slurry infiltration process is an essential factor affecting slurry diffusion [15–17]. The main research methods include four [18]: a macro model based on macro experimental phenomena, the study of the analysis method of single particle trajectory of suspension, the method of studying congestion through probability, and the mesh model method. The research on the filtering effect in soil mass infiltration grouting is developed based on the research of other disciplines. The filtering effect has been confirmed in the uniaxial sand soil infiltration grouting test [19, 20]: the test shows that, as the filtered cement particles gradually block the soil mass pores, the grouting pressure gradually increases. At the same time, the concentration of grout injected into the soil mass is uneven, and the closer to the grouting mouth, the greater the concentration of grout. These test results show the importance of considering the filtration effect. Currently, there is relatively little research on the filtration effect in seepage grouting. Some scholars have conducted indoor tests to study the seepage effect's influence on grouting diffusion [21–23]. In terms of model, F Bouchelaghem [24] studied the micro mechanism of the filtration effect, the coupling calculation model of fluid convection, hydrodynamic dispersion and other factors. However, because the mechanism of the seepage effect is very complex, it is impossible to obtain the change rule of a physical quantity in the process of mud diffusion comprehensively and intuitively through experimental methods. In terms of numerical calculation, some scholars have established iterative numerical calculation methods to consider the seepage effect from the perspective of mass conservation [25]. Or use the grid model method to study the sand column grouting model considering the seepage effect [26]. J. S. Kim et al. [27] gave an iterative numerical calculation method combining mass conservation equation and filtering law. S Maghous et al. [28] extended it to column infiltration grouting. The above research methods all consider the filtration effect from the perspective of quality conversion and have their advantages. However, most existing models are one-dimensional models based on indoor one-way grouting tests, which cannot be directly applied to on-site grouting. There is a lack of research on the fluidity characteristics of coal gangue slurry under the effect of seepage in goaf. Therefore, it is necessary to carry out an experimental study on seepage grouting of coal gangue slurry in the pore medium of goaf.

Therefore, based on the Darcy seepage law, a seepage theoretical calculation model considering the time-space effect of seepage is established, the "water-cement ratio change matrix" in the seepage process of coal gangue slurry is deduced, and the slurry particle deposition law is analyzed. The self-developed visual testing platform for mud diffusion in goaf is used for testing. The AHFO is used to monitor the flow and diffusion of coal gangue slurry in the collapse area of the goaf, and the gravity gradient and water-cement ratio of the slurry in the goaf are measured. Finally, carry out an engineering test.

## 2 Methodology

### 2.1 Gangue slurry filling technology in goaf

Gangue slurry pipeline filling technology refers to the coal-based solid waste produced in the production process of coal mines or power plants made into powder with a certain particle size by crushing, grinding and other technical means and then mixed with water in a specific proportion to form a slurry. Then, the gangue slurry is transported to the gap between rock

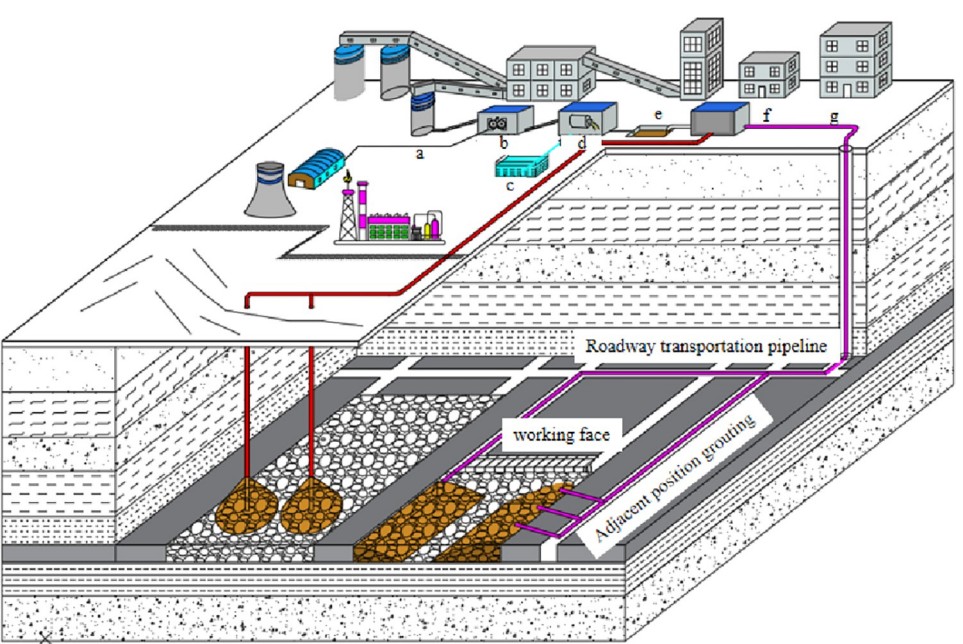

**Fig 1. Pipe filling of gangue slurry in goaf.** a-Gangue bin, b-Crushing station, c-Reservoir, d-Grinding station, e-Collecting tank, f-pumping station, g-Transmission pipeline.

blocks in the collapse zone behind the working face by the grouting pump through the ground transmission pipeline, blanking drilling hole, main roadway transmission pipeline and low-level grouting pipe. Finally, the harmless disposal of gangue can be realized without affecting the normal production of the working face.

The main technical means of filling are adjacent grouting filling, and its technical schematic diagram is shown in Fig 1. This technology has little impact on the original production system, can realize the parallel operation of mining and filling, the transportation mode of filling materials is efficient and fast, and the filling process can realize intelligent control. It is an effective technical way for the green disposal of gangue, as shown in Fig 1.

Adjacency grouting and filling refers to the arrangement of adjacency grouting and filling boreholes in the upper area of the collapse zone and the construction of slurry filling channels in the inclined space of the same layer. Its implementation means mainly drilling upward from the adjacent working face roadway or adjacent main roadway to the upper area of the collapse zone. The conditions of adjacency grouting filling are relatively wide, which can be applied to the working face under mining to realize filling with mining. At the same time, it can also be used as a technical means of slurry filling in the old goaf by constructing upward drilling in the main roadway or stopping the line to realize post-mining filling. The coal gangue and water are mixed into a particular concentration of gangue slurry through the underground pulping system. The adjacent grouting form is used to fill the collapse zone of the goaf to realize the harmless disposal of coal gangue under low interference in the working face. Due to the different filling purposes, the gangue slurry filling is different from paste filling, and the strength of the filling is not too high, mainly through optimizing the gangue slurry ratio and technology to increase the amount of gangue filling. The content of fine-grained aggregate decreased, resulting in significant differences in flow performance between coal gangue slurry and traditional paste [29–31]. At the same time, to reduce the filling cost, the addition of cementitious materials was canceled.

## 2.2 Analysis of influence mechanism of space-time effect of grouting filtration

The percolation effect refers to that in the process of slurry infiltration, due to the filtration of porous media, the slurry particles deviate from their trajectory and constantly stagnate, precipitate, adsorb and deposit. And then occupy space to reduce the pore opening, change the permeability of porous media, and finally block the permeability channel, causing the change of slurry water-cement ratio, which has an obvious space-time effect.

The percolation effect widely exists in the process of suspension seepage in porous media. When the slurry diffuses in porous media, part of the cement particles will be "blocked" by the sand particle skeleton, making the solid particles stay between the pores of the sand, causing the siltation and blockage of cement particles in the sand layer. This phenomenon leads to the reduction of the ability of cement slurry to penetrate sandy strata, which has been identified as the main factor in controlling slurry infiltration grouting in many studies [32, 33]. The traditional slurry seepage theoretical model only considers the variable $t$ and has not considered the influence of the diffusion distance, which may lead to the deviation of the calculation results of the theoretical model. Therefore, for seepage grouting in porous media, it is necessary to consider the relationship between the seepage coefficient and grouting time and diffusion distance.

Based on Darcy's law, according to the equivalent relationship between seepage flow and pore volume, a theoretical calculation model of seepage considering the time-space effect of seepage is established [34]:

$$dr_1 dQ = (H - h_0)Ak(r_1, t)dt \tag{1}$$

Where $H$ is the grouting pressure head, $h_0$ is the pore pressure head, $r_1$ is the grouting radius, $A$ is the sectional area of the grouting pipe.

According to the foregoing analysis, $k(r_1,t)$ is the function of the seepage coefficient of the slurry varying with the grouting time and diffusion distance.

$$k(r_1, t) = \frac{f(t)}{f(r_1)} \tag{2}$$

Substitute Formula (2) into Formula (1) and integrate it to obtain the grouting volume Q.

$$\begin{cases} Q = \dfrac{(H - h_0)A \displaystyle\int_0^t f(t)\mathrm{d}t}{\displaystyle\int_{r_0}^{r_1} f(r_1)\mathrm{d}r_1} = A(r_1 - r_0) \cdot n \\[4ex] \dfrac{\displaystyle\int_0^t f(t)\mathrm{d}t}{\displaystyle\int_{r_0}^{r_1} f(r_1)\mathrm{d}r_1} = \dfrac{(r_1 - r_0) \cdot n}{H - h_0} \end{cases} \tag{3}$$

Where $r_0$ is the radius of the grouting pipe, $r_0 \ll r_1$.

From Eq (3), it can be seen that if the expressions of functions $f(t)$ and $f(r_1)$ are known, they can be solved, and these two expressions can only be obtained through a large number of indoor test analyses. Combining Eq (2) and Eq (3), obtaining $k(r_1,t)$ is equivalent to obtaining

the seepage coefficient $K(r_1,t)$ and porosity $n(r_1,t)$ of rock and soil mass that is [35]

$$K(r_1, t) = 2\left(\frac{n(r_1, t)}{1 - n(r_1, t)}\right)^2 d_{10}^2 \tag{4}$$

Due to the influence of the seepage effect, the seepage coefficient of rock and soil mass changes with time, mainly reflected in the change of porosity, as shown in the following equation [36, 37]:

$$\begin{cases} \sigma = \beta\alpha\omega t V \\ n(t) = n - \beta\alpha\omega t \end{cases} \tag{5}$$

Where $\sigma$ is the volume of slurry particles percolated in $t$ time, $\beta$ is the expansion coefficient of particles, generally taken as 2.0~3.0. $\alpha$ is the percolation coefficient of the slurry, and its value is related to the filtration rate and filtration coefficient, $\omega$ is the mass fraction of particles in the slurry, $V$ is the total volume of rock and soil mass, $n(t)$ is the real-time porosity of rock and soil mass.

The percolation process is divided into two parts. The formation of slurry particle percolation takes a certain time, and the filtration effect has not been completed. After the slurry percolates for a certain time, the slurry particles complete the percolation, and then the filtration starts. Therefore, the complete percolation process is the circulation process of slurry infiltration and percolation.

The increased slurry particle precipitation directly leads to decreased porosity and seepage coefficient. When the pore volume is certain, the change in slurry particle precipitation quantity represents the change in the water-cement ratio. Therefore, the slurry water-cement ratio change is an important embodiment of the filtration effect. Now the water-cement ratio is taken as the research object to analyze its change process and characteristics. Since the slurry infiltration is a step-by-step process, for the convenience of expression, the matrix form is used to describe and establish the "water cement ratio change matrix" of the slurry infiltration process, as shown in the following Eq (6).

$$\mathbf{R}_{ij} = \begin{bmatrix} r_{11} & \cdots & \cdots & \cdots & \\ r_{11} - \Delta r_{21} & r_{11} + \Delta r_{11} & \cdots & \cdots & \cdots \\ r_{11} - \Delta r_{21} - \Delta r_{31} & r_{11} + \Delta r_{11} - \Delta r_{32} & r_{11} + \Delta r_{11} + \Delta r_{22} & \cdots & \cdots \\ r_{11} - \Delta r_{21} - \Delta r_{31} - \Delta r_{41} & r_{11} + \Delta r_{11} - \Delta r_{32} - \Delta r_{42} & r_{11} + \Delta r_{11} + \Delta r_{22} - \Delta r_{43} & r_{11} + \Delta r_{11} + \Delta r_{22} + \Delta r_{33} & \cdots \\ \cdots & \cdots & \cdots & \cdots & \cdots \end{bmatrix} \tag{6}$$

Where $i$ and $j$ are the unit length of grouting seepage distance and the unit length of the injected medium, $i \geq 1$, $j \geq 1$. It can be seen from the above formula that $r_{11}$ is the initial water-cement ratio of the slurry, and $r_{i1} \leq r_{11}$, $r_{ij} \leq r_{i(j+1)} \leq r_{i(j+2)} \leq \ldots \leq r_{i(j+m)} \leq \ldots$, $m \geq 1$. The water-cement ratio of the grout increases along the seepage direction after the grout passes through the percolation. The farther away from the grouting pipe, the greater the water-cement ratio of the grout. It is especially pointed out that the water-cement ratio near the grouting mouth is smaller than the initial water-cement ratio.

The filtration effect has an obvious space-time effect, which is not only related to percolation time but also related to diffusion distance. In the direction perpendicular to the seepage, the seepage coefficient decreases with time; In the direction parallel to the infiltration direction, the water-cement ratio gradually increases with the diffusion distance. The water-cement ratio near the grouting hole is less than the initial water-cement ratio, resulting in the

contradiction of a small water-cement ratio at the near end and a large water-cement ratio at the far end. Horizontally, the amount of slurry particle precipitation increases with time; Vertically, the amount of slurry particle precipitation changes with the diffusion distance and decreases relatively in turn. It is consistent with the analysis of test results in subsequent chapters.

## 3 Goaf coal gangue slurry diffusion test platform

The model test is widely used in various scientific research fields as a traditional test method. For common engineering problems, such as the inability to determine the row spacing between grouting holes on site, the solution must be obtained in the laboratory using model tests. This shows the importance of the model test. The temperature of the gangue slurry diffusion process in the model is monitored through and in combination with optical fiber sensing technology. The gangue slurry diffusion range is characterized by the software interpolation method to realize the "visualization" of the gangue slurry diffusion morphology in the model.

The grouting simulation system assembled indoors is adopted. The system comprises three parts: pressure supply, test box, and optical fiber monitoring. The simulation system is shown in Fig 2. The pressure supply system comprises an air compressor, slurry storage barrel, grouting pipeline, electronic scale, pressure gauge, and pressure regulating valve, which can provide stable pressure. The test chamber comprises a detachable acrylic box with a side length of 50 cm. Gangue slurry diffusion can be observed on the front of the box. The grouting pressure of this test is 0.032 MPa. The monitoring system comprises an active heating optical fiber connected with a DTS temperature demodulator and a DC heating box. The DTS can realize active heating and measure the temperature along the optical fiber with high accuracy, which can be well applied to this test. Since the active heating optical fiber mainly realizes the visualization of the gangue slurry diffusion range by monitoring the temperature change after the gangue slurry injection, it is unnecessary to calibrate the thermal conductivity of the active heating optical fiber and the simulated goaf gangue.

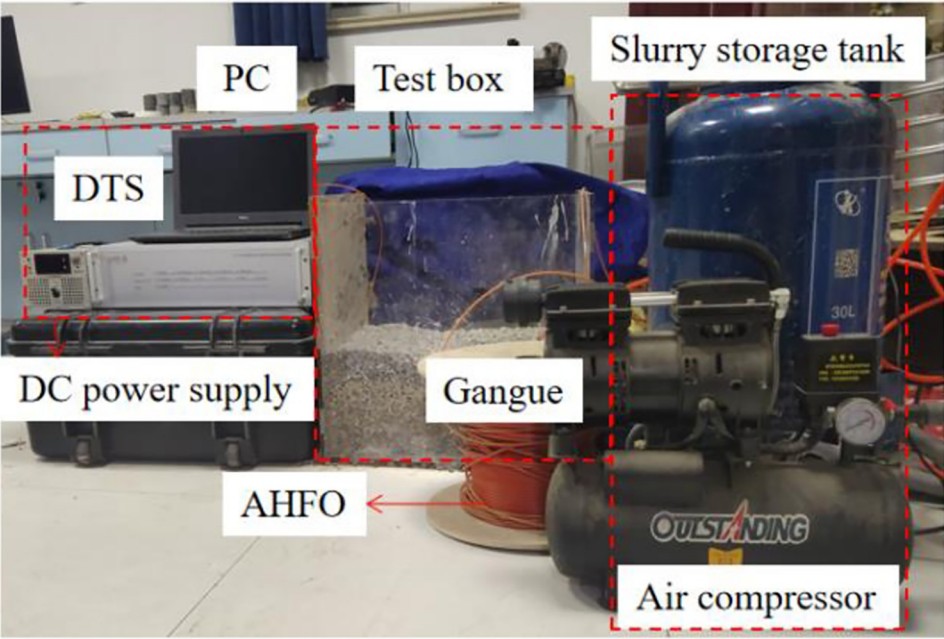

**Fig 2. Test platform for coal gangue slurry diffusion in goaf.**

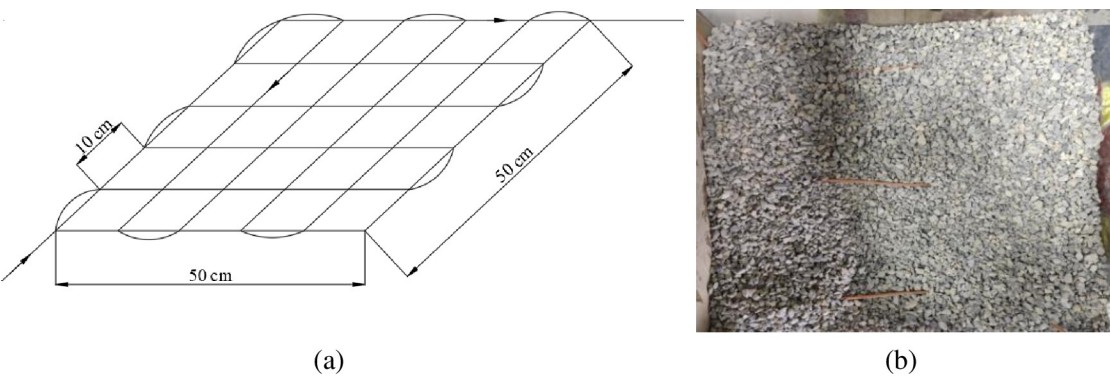

(a)                                                          (b)

**Fig 3. Arrangement method of active heating optical fiber.**

As shown in Fig 3(A), in building the model from bottom to top, the optical fiber shall be laid at a position 5 cm away from the bottom plate of the model [38]. First, it shall be laid longitudinally 6 times with a spacing of 10 cm and then laid horizontally to form a square grid. After laying the lower layer, continue to fill the gravel with a filling height of 20 cm. As shown in Fig 3(B), the optical cable is buried in the gangue pile in the goaf. The grid laying method can improve the spatial resolution of optical fiber monitoring and monitor the gangue slurry diffusion range.

As shown in Fig 4, firstly, 17 ~ 20 m, 22 ~ 24 m, 26 ~ 29 m, 36 ~ 38 m and 35 ~ 36 m sections of 100 m carbon fiber heating optical fiber are placed in the temperature control room respectively to maintain the temperature at 30°C, and then repeated tests are carried out. After five consecutive measurements, the relationship diagram between the optical fiber distribution can be made. When analyzing the data obtained from repeated tests, the standard deviation of the measured data can be used as the measurement standard. According to Fig 5(A), the five groups of data measured are basically the same. According to Fig 5(B), the standard deviation of the data is less than 0.25, which shows that the DTS thermometer has good repeatability.

## 4 Flow performance of gangue slurry

### 4.1 Basic characteristics of gangue slurry

**4.1.1 Basic physical and chemical properties of gangue powder.** The X-ray diffractometer tested the mineral composition of gangue filling material. According to the standard

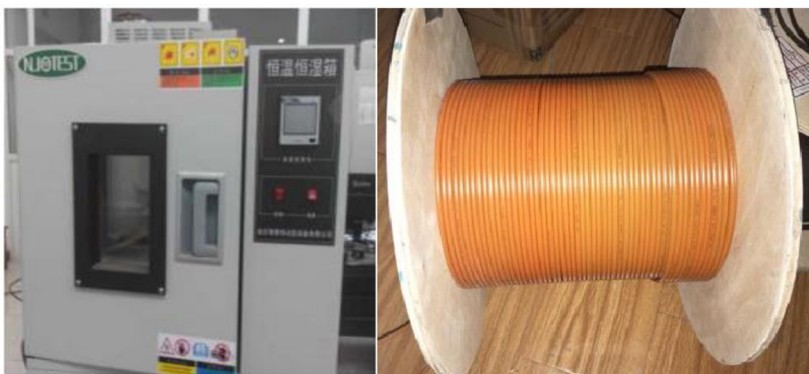

(a) Constant temperature and humidity instrument                    (b) AHFO

**Fig 4. Optical fiber repeatability test.** (a) Constant temperature and humidity instrument. (b) AHFO.

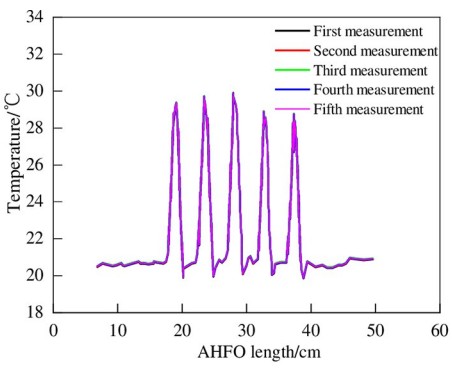
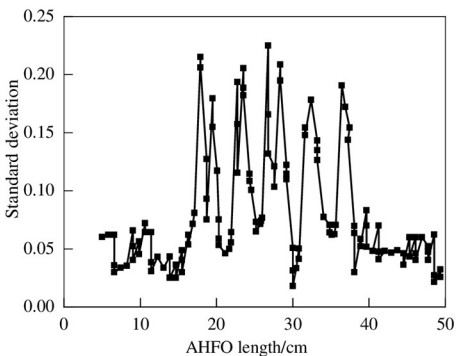

(a) Repeatability test results

(b) Standard deviation of measurement data

**Fig 5. Repeatability test results of optical fiber.** (a) Repeatability test results. (b) Standard deviation of measurement data.

powder diffraction data of various substances provided by the national data centre of the powder diffraction Federation (TCPDS-ICDD), the comparative analysis was carried out according to the standard analysis method. The analysis results are as follows: the phase composition mainly includes quartz, kaolinite, some feldspar, mica and a small amount of chlorite, calcite, siderite, pyrite and other minerals. In addition, the main chemical components are shown in Table 1.

**4.1.2 Particle size distribution of gangue.** The crushed gangue powder is the main aggregate of gangue slurry filling material. According to laboratory measurement, the particle size range of gangue powder used in this test is 2.50 ~3.0 mm and 2.00 ~2.50 mm, 1.50 ~2.00 mm, 0.50 ~1.00 mm, <500 μm. The proportions of these are 3.88%, 2.83%, 5.95%, 9.93%, 27.98% and 44.03% respectively. The particle size distribution curve is shown in Fig 6.

**4.1.3 Slump test of gangue slurry.** According to the basic test results, the mass concentration of the gangue slurry selected for the test is 70%. According to the existing practical experience of paste filling, the slump of the slurry that meets the pumping is at least 15~20 cm. According to the practical experience of the Jinchuan nickel mine, Tonglushan Copper Mine and other mines, the slurry has good fluidity when the slump is 25~28 cm. The slump test of coal gangue slurry with 70% concentration is carried out. The results of slump and expansion obtained from the test are shown in Table 2. The average gravity gradient of the gangue slurry is calculated to be 3.8%.

## 4.2 Filtration time-space effect of gangue slurry flow process

Due to the influence of the filtration effect, particle settlement may occur in the gangue slurry during the flow process. After the test, evenly sample the two points 5 cm and 45 cm away from the grouting hole boundary, as shown in Fig 7. After sampling, the slurry is dried and screened. The proportion of gangue powder less than 200 mesh after drying is 9.81% and 20.36%, respectively, and the water-cement ratio of gangue slurry at the corresponding measuring point 1 is small; The water cement ratio of gangue slurry at measuring point 2 is large.

**Table 1. Main chemical composition of coal gangue.**

| Composition | $SiO_2$ | $Al_2O_3$ | $Fe_2O_3$ | CaO | MgO | $K_2O$ | $Na_2O$ | $SO_3$ | Other |
|---|---|---|---|---|---|---|---|---|---|
| Mass fraction /% | 50.83 | 26.344 | 6.97 | 3.54 | 1.36 | 0.84 | 0.56 | 0.23 | 8.53 |

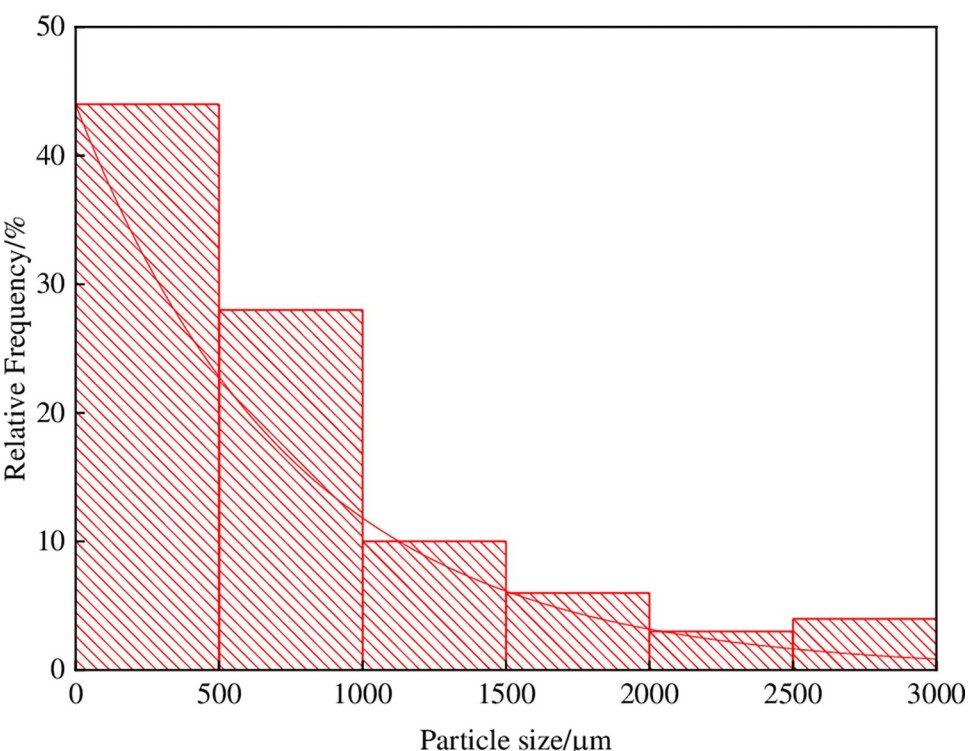

**Fig 6. Particle size distribution of crushed gangue powder.**

In the direction parallel to the infiltration direction, the water-cement ratio gradually increases with the diffusion distance. The water-cement ratio near the grouting hole is less than the initial water-cement ratio, the near-end water-cement ratio is small, and the far-end water-cement ratio is large. The experimental results also verify the correctness of the theoretical analysis.

According to the screening results, the proportion of fine particles smaller than 200 mesh in the flow direction of the gangue slurry has increased. The proportion of coarse particles larger than 200 mesh in the corresponding flow direction has decreased, indicating that the solid particles in the gangue slurry appear to be a particle sorting phenomenon due to the influence of the seepage effect in the flow process of the goaf. The main reasons for this phenomenon are: the slurry is mainly distributed in the cavities composed of each rock block and the voids between each rock block. The slurry in different cavities is in a polarized state. When the cavity is filled with slurry, the voids between rock blocks around the cavity are filled with slurry, but there are still some small voids without a slurry. When there is no slurry in the cavity, the slurry in the interstices between the rock blocks around the cavity is distributed as a broken branch. That is, the slurry blocks the crevices between the rock blocks (there are large

**Table 2. Test results of slump and expansion.**

| Number | Mass concentration | Slump (cm) | | Expansion (cm) | |
|---|---|---|---|---|---|
| | | Measured | Mean | Measured | Mean |
| **1** | 70% | 28.2 | 28.1 | 50.4 | 50.1 |
| **2** | | 27.6 | | 49.7 | |
| **3** | | 28.1 | | 50.1 | |

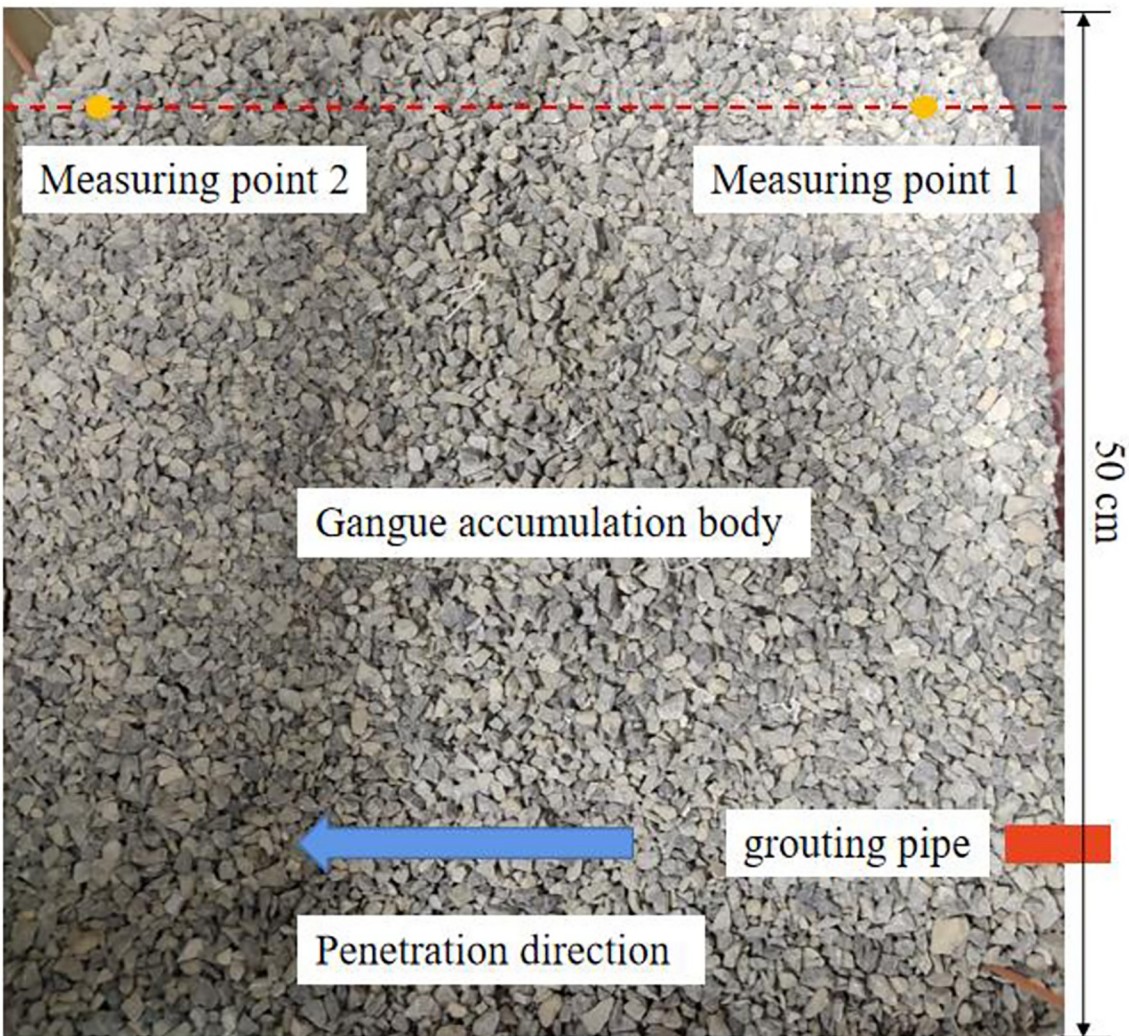

**Fig 7. Gangue slurry sampling after filling test.**

particles in the slurry or the crevices between the rock blocks are too small), indicating that the slurry flow in the interstices between the rock blocks is selective. The space shapes of voids in the gangue pile are different and randomly distributed. When small pores appear locally, the coal gangue slurry flow process, velocity, size and direction often change significantly. Finally, the flow state of the gangue slurry changes greatly, and the stability decreases. Finally, the settling velocity of coarse particles in the coal gangue slurry increases along the vertical direction. At the same time, the phenomenon of large particle aggregation and sedimentation can also be observed where the gap of local gangue accumulation is small. The coarse particles with the settlement are mostly 2.0~3.0 mm, which shows that the gap with a small size in the goaf is the main position for separating gangue slurry particles, and mainly acts on the coarse particles with the size of 2.0 mm~3.0 mm. Based on this, the particle gradation in the gangue slurry filling process needs to be reasonably considered to avoid the large proportion of coarse particles, resulting in the dense settlement at local positions in the flow process of gangue slurry goaf. The test results show that the slurry shows a significant percolation effect in the direction of seepage diffusion. The farther away from the grouting pipe, the greater the water-cement ratio

of the slurry, which corresponds to the reduction of the proportion of coarse particles, which is consistent with the theoretical analysis results.

## 4.3 Flow law of gangue slurry

During the filling process, the digital camera is used to record and observe the mud flow state through the transparent acrylic plate. According to the observation of the test process, the coal gangue slurry can flow through the void space of the accumulation, and the overall fluidity is good. The overall flow law shows the law of first longitudinal expansion and then horizontal diffusion. At the early stage of the test, the coal gangue slurry first formed a semicircular over-burden with a radius of about 15 cm on the coal gangue pile under the grouting hole. With the continuous increase of the filling amount, the gangue slurry will not expand further along the upper boundary of the vein accumulation body. However, it extends to the lower part of the coverage area, forming a longitudinal expansion channel within a certain range near the ori-fice. When the mud diffuses to the bottom boundary of the tank along the longitudinal chan-nel, the mud starts to spread laterally and forms a certain slope along the diffusion direction. As the filling volume increases, the gangue slurry spreads laterally until it contacts the bound-ary of the tank. After contacting the boundary, the height of the gangue slurry continues to increase with the filling volume. However, the liquid level gradient remains stable, and the flow process is shown in Fig 8.

Using the monitoring results, the stacking height curves when the filling volume is 4 L, 6 L and 8 L are drawn, as shown in Fig 9. The gravity flow gradient near the grouting hole is signif-icantly greater than that in the area far away from the grouting hole. This is because the main flow of gangue slurry near the grouting hole is longitudinal expansion, so there is accumula-tion at the bottom of the box near the grouting hole, with a large slope. During the accumula-tion process, the gangue slurry diffuses laterally by gravity and gravity.

With the increased distance from the grouting hole, the gravity flow gradient of the gangue slurry first decreases and finally becomes stable. According to the variation law, the gravity flow slope of the gangue slurry in the gangue accumulation body can be roughly divided into

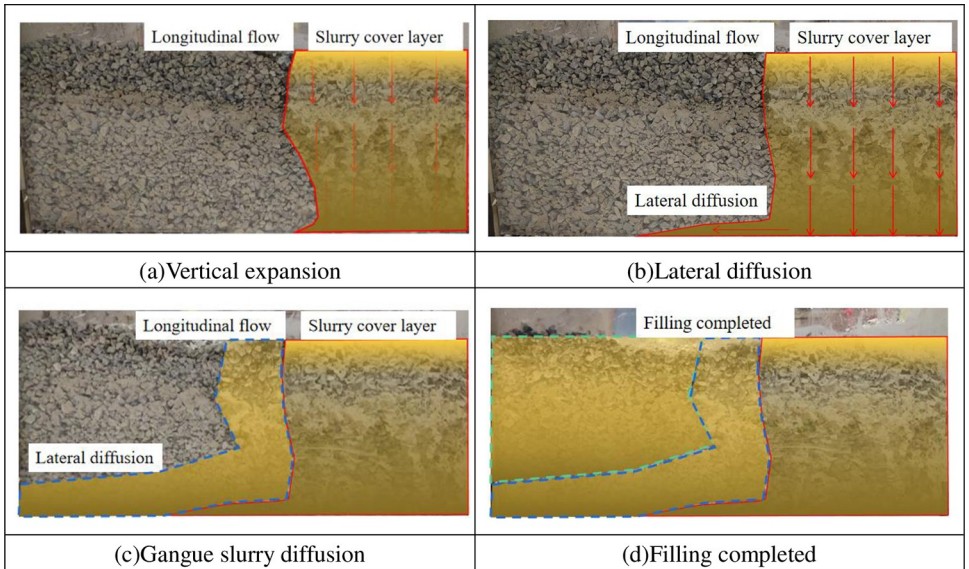

**Fig 8. Flow process of gangue slurry in gangue accumulation.** (a) Vertical expansion. (b) Lateral diffusion. (c) Gangue slurry diffusion. (d) Filling completed.

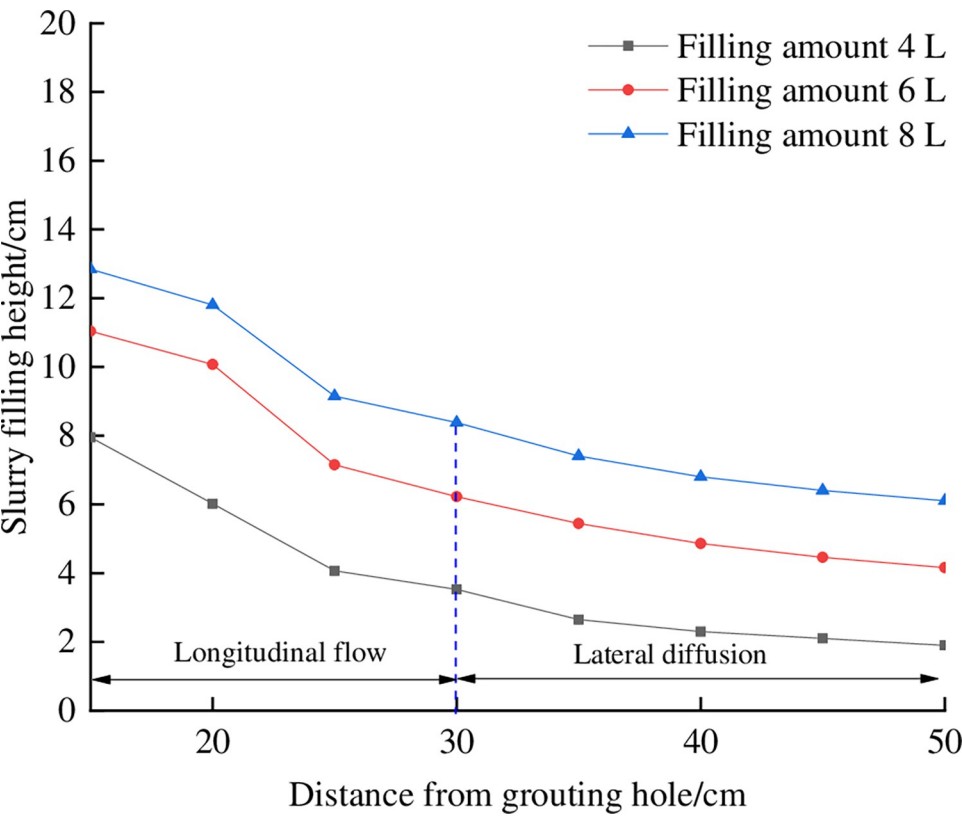

**Fig 9. Curve of gangue slurry stacking height.**

two areas. The longitudinal flow class area is within the range of 0~30 cm from the grouting hole. Affected by the longitudinal expansion area, the gravity flow slope is relatively large, with an average value of 17.62%; In the lateral diffusion class area, the gravity gradient tends to be stable within 30~50 cm from the grouting hole, with an average value of 6.34%.

The slope of the gangue slurry after stabilization in the gangue stack is 6.34%, while the slope of the gangue slurry in the slump test is 3.8%, which is higher than the gravity slope in the gangue stack in the slump test. This is because the gangue accumulation body has a certain blocking effect on the gangue slurry flow. At the same time, the surface of the gangue accumulation body is relatively dry, which absorbs part of the water in the gangue slurry during the gangue slurry flow, increasing the gangue slurry concentration, further weakening the lateral diffusion ability of the gangue slurry and increasing the gravity flow slope.

## 4.4 AHFO monitoring grouting diffusion range

After grouting, the DTS demodulator obtains the temperature along the AHFO line. According to the two-dimensional positioning algorithm, combined with the grid parameters of the optical fiber deployed in the experiment, the corresponding coordinates of each sampling point on the optical fiber are obtained. Using the Kriging interpolation method introduced, the temperature of the AHFO sampling point is regarded as a known point, and the temperature value of unknown points in the monitoring range is estimated to obtain the temperature field distribution of the model bottom plate.

As shown in Fig 10(A), a three-dimensional coordinate system is established with the starting position of the optical fiber as the origin, the transverse length of the grid as the X-axis, the

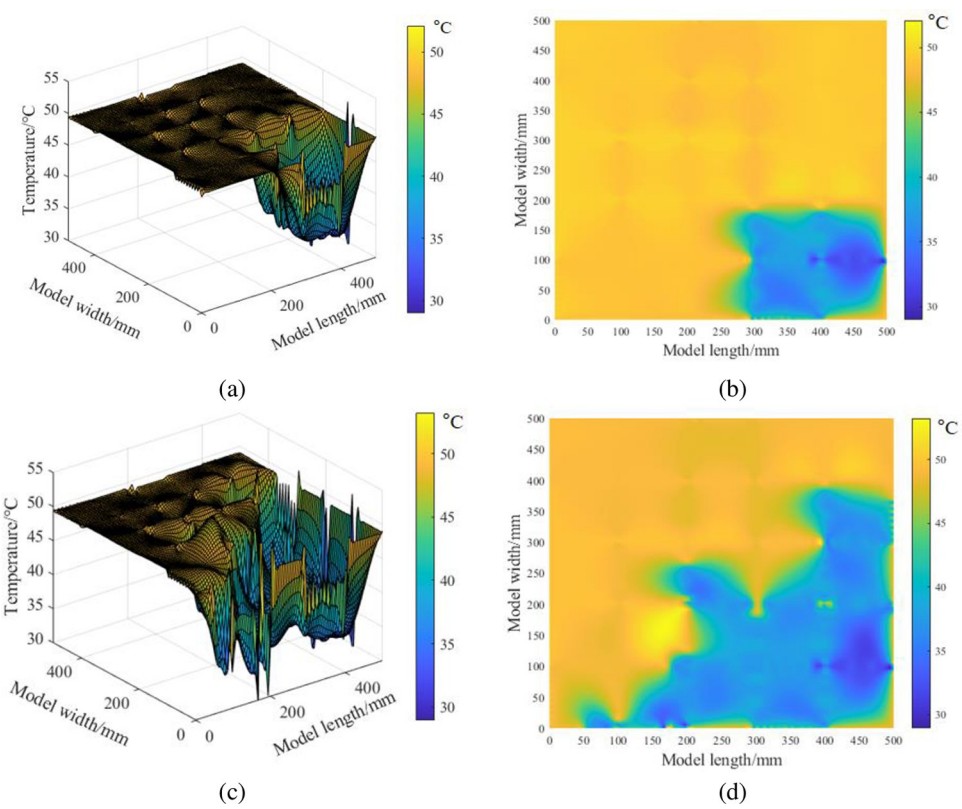

**Fig 10. Grouting diffusion range.**

longitudinal length of the grid as the Y-axis, and the monitored temperature as the Z-axis. The temperature field with the grouting pipe as the center is the lowest and gradually radiates around. As shown in Fig 10(B), the grouting diffusion temperature field is shown. In the lower right corner of the model, an irregular elliptical grouting area is formed with the grouting pipe as the center, which corresponds to the longitudinal expansion stage of the grout. As shown in Fig 10(C), this is the three-dimensional temperature distribution of gangue slurry diffusion. As shown in Fig 10(D), the gangue slurry first expands to both the model's sides and the centre to form a lateral seepage channel. This stage corresponds to the transverse seepage stage in the above chapters.

## 5 Application of adjacent grouting filling of gangue slurry

### 5.1 Project overview

The raw coal production capacity of the well is 10 Mt/a, and the total amount of gangue is about 0.85 Mt/a producing the pure gangue produced bottom raising and chamber excavation during the layout of the underground working face is about 20 Mt/a.

### 5.2 Filling scheme

According to the preliminary test results, it is determined that the mass concentration of gangue slurry in the project application stage is 70%, the pumping pressure is 8 Mpa, the working flow rate is 1.5 m/s, the selected plunger filling pump model is HBMG30/21-110S, and the selected mixer model is JSY3200. The process is divided into the following steps: quantitative feeding → mixing and pulping → pumping → filling.

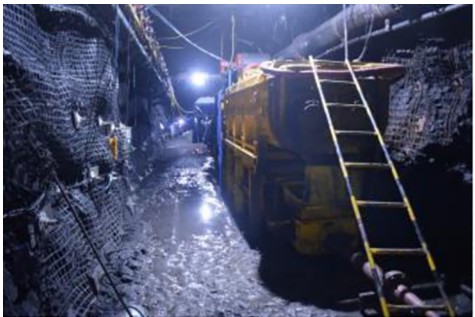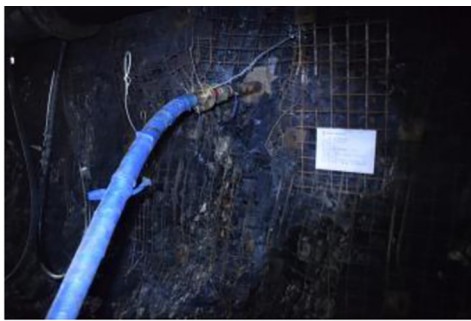

(a) Pulping station    (b) Filling borehole

**Fig 11. Site of adjacent grouting filling.** (a) Pulping station. (b) Filling borehole.

Firstly, the crushing, pulping and filling equipment is arranged in the machine head chamber of working face 301 by using the abandoned roadway of the mine. The underground raw gangue is crushed in two stages to form a powder with a particle size of less than 3 mm. Then it is made into 70% slurry by adding the existing underground mine water through a quantitative feeder. Then it is transported to the adjacent borehole with a plunger pump and pipeline for filling, as shown in Fig 11.

Based on the theoretical calculation, measured data, and the analysis of the roof condition of the working face, the height of the gob collapse zone formed after the coal seam mining in working face 301 is 9.2~16.9 m. Therefore, to ensure that the final hole is located in the collapse zone, the final hole height of the filling borehole is 11m, the elevation angle is 9°, and the borehole diameter is designed in the early stage 133 mm. The protective casing is placed in the whole process of grouting drilling to isolate the gangue from the gob. The end of the grouting pipe is 3.0~5.0 m connected with the flower pipe to increase the grouting range, and the hole diameter of the flower hole is 25 mm.

## 5.3 Filling effect

The on-site pressure monitoring data shows that the pressure of the pipeline and filling borehole is stable during the grouting period, which can realize continuous and stable filling operation, and the grouting and filling effect adjacent to the goaf is good. The feasibility of the underground adjacent grouting filling technology is successfully verified, which further proves that there is sufficient filling space in the gob collapse zone and that the equipment and transportation parameters selected for engineering application are reasonable, and the system stability is high.

During the filling period, the transportation state of the gangue slurry pipeline is stable. The solid particles in the slurry have not silted up, the slurry has good fluidity in the void space of the goaf, and there is no aggregation and settlement phenomenon, indicating the transportation flow rate and slurry grading concentration of the slurry pipeline designed in the early stage is reasonable.

When the cumulative grouting volume of a single hole is 700 m$^3$, the grout diffusion distance is expected to be 51 m according to the gravity flow gradient of the grout in the goaf of 6.34%. Pass through the inclined hole of coal pillar construction at the position 45 m away from the borehole, and the hole depth is 36m. Ensure that the final hole of the borehole is located above the bottom plate. Immediately after the hole is formed, use the borehole peep to

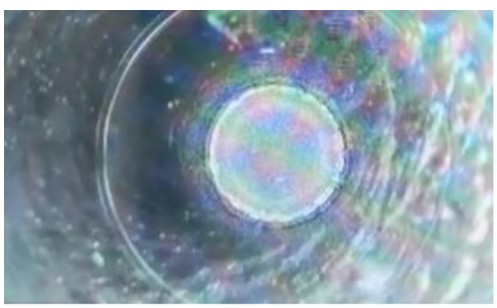 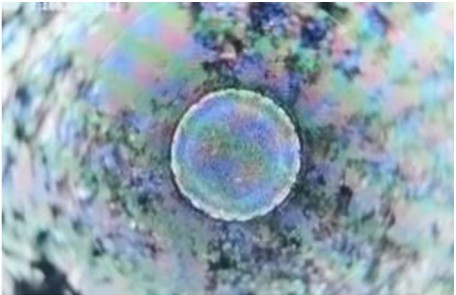

(a) Gangue free slurry          (b) Gangue slurry

**Fig 12. Bored peeking pictures.** (a) Gangue free slurry. (b) Gangue slurry.

peep at the situation in the hole. The peeping situation in the hole is shown in Fig 12. The results of borehole peeping show that the lens of the peeping mirror is blocked by the slurry, indicating that the coal gangue slurry has spread to this position. The filling slurry has good flow and diffusion performance in the goaf, and the flow law conforms to the test results. The slurry diffusion distance is greater than 45 m. The accuracy of slurry seepage slope in goaf obtained from laboratory tests is verified.

## 6 Conclusion

1. Through theoretical analysis, the "water-cement ratio change matrix" in the coal gangue slurry seepage process is deduced to reveal the basic mechanism of the porous media filtration effect, and the water-cement ratio gradually increases in the seepage process of gangue slurry.

2. The overall flow law of the gangue slurry in the goaf shows the law of first longitudinal expansion and then horizontal diffusion. Before the slurry is stabilized, the seepage gradient is large, and after stabilization, the seepage gradient decreases, the average seepage gradient is 6.34%.

3. Filtration effect has an obvious space-time effect. The proportion of gangue particles less than 200 mesh in the gangue slurry near the grouting mouth is 9.81%, while the far end position is 20.36%. In the direction parallel to the infiltration direction, the water-cement ratio gradually increases with the diffusion distance. The farther away from the grouting mouth, the greater the water-cement ratio of the slurry.

4. According to the permeability gradient obtained from the test and the height of the grouting pipe, the grouting diffusion distance is predicted to be 51m. The engineering application results show that the cumulative grouting volume of a single hole is 700 $m^3$, the slurry diffusion distance is greater than 45 m, and the slurry has excellent fluidity in the void space of the goaf.

## Acknowledgments

Many thanks to Professor Chai for the guidance of this paper. The author would also like to thank Dr. Zhu and Dr. Gu for his valuable comments and suggestions for improvement of the manuscript.

## Author Contributions

**Data curation:** Yibo Ouyang.

**Methodology:** Wenzhe Gu.

**Validation:** Fengqi Qiu.

**Writing – original draft:** Lei Zhu.

**Writing – review & editing:** Yibo Ouyang.

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
