## [Decision Letter · Decision Letter 0]

26 Sep 2022

PONE-D-22-23513Grouting mechanism and experimental study of goaf considering percolation effectPLOS ONE

Dear Dr. GU,

Thank you for submitting your manuscript to PLOS ONE. After careful consideration, we feel that it has merit but does not fully meet PLOS ONE’s publication criteria as it currently stands. Therefore, we invite you to submit a revised version of the manuscript that addresses the points raised during the review process.

We look forward to receiving your revised manuscript.

Kind regards,

Zhiwei Peng, Ph.D.

Academic Editor

PLOS ONE

Journal Requirements:

- The name of the colleague or the details of the professional service that edited your manuscript.

- A copy of your manuscript showing your changes by either highlighting them or using track changes (uploaded as a *supporting information* file).

- A clean copy of the edited manuscript (uploaded as the new *manuscript* file)”.

Reviewers' comments:

Reviewer's Responses to Questions

**Comments to the Author**

1. Is the manuscript technically sound, and do the data support the conclusions?

Reviewer #1: Partly

Reviewer #2: Yes

2. Has the statistical analysis been performed appropriately and rigorously? 

Reviewer #1: N/A

Reviewer #2: Yes

3. Have the authors made all data underlying the findings in their manuscript fully available?

Reviewer #1: Yes

Reviewer #2: No

4. Is the manuscript presented in an intelligible fashion and written in standard English?

Reviewer #1: No

Reviewer #2: Yes

5. Review Comments to the Author

Reviewer #1: 1、This article is not written in standard English, and the language needs to be improved greatly.

2、the first paragraph in Introduction seems to be irrelevant to this study

3、What does "three under" coal mining mean in the first paragraph?

4、It's very confusing to talk about slurry filling and gangue filling in the article. Please clarify the relationship and what is the main research object

5、Line 57: “At present, scholars at home and abroad have put forward several grouting diffusion theories through assumptions and simplified methods, including penetration grouting, splitting grouting and compaction grouting, of which penetration grouting is the most commonly used.” Is there any literature basis for this statement, why is it more widely used, and whether it is more in line with engineering practice? Please explain the reason.

6、Starting from line 63, references frequently use “It...” This is not a standard writing format, and the overall description of the paragraph is illogical. please revise carefully

7、Introduction was written in a confused way. It did not describe and summarize the current situation of grouting research. It was impossible to see what was better than the existing research or what was considered more or what problems were solved

8、Line 129: “which has been identified as the main factor in controlling slurry infiltration grouting in many studies” Please list relevant research literature

9、Please analyze in detail what each item in formula (7) expresses and how to get the subsequent conclusion from this formula

10、How to verify that the established Goaf coal gangue slurry diffusion test platform is similar to the actual goaf situation, and how to prove that it can represent the actual grouting situation?

11、In Section 4.4, how many sampling points are there in the experiment? Please show the location of the sampling points with a diagram. Can the number meet the requirements of Kriging interpolation?

12、The grouting pressure is an important factor, but it is not described in Section 4.3

13、The display of flow process of slurry in Figure 8 is too little to show the specific situation

14、What do the various colors in Figure 10 mean?

Reviewer #2: The submission investigates the grouting mechanism and experimental study of goaf. It is an interesting work which is a hot research topic recently. In general, this work was well prepared but the English language need to be polished by a native English speaker. Although the paper presents experimental results that are potentially useful, the study has several limitations. The limitations, in my opinion, are as follows:

1) The abstract should be polished and stress the main points.

2) The presentation level is not adequate, especially the Introduction. The outcomes of the past researches and their relation to the present study should be clearly presented. At present, the Introduction seems like a string of studies that were done on the topic rather than a meaningful narrative.

3) In line 32, AHFO is not suitable to be one key word.

4) It is better to reconsider the title of each section.

5) How to reflect the field performance by the results of test platform? Is there any difference?

6) Currently, the Conclusion presents a relatively verbose summary of the results – I recommend making the results more general and especially pointing out the practical implications of the research.

7) There have numerous format problems, eg. always missing a gap between a number and unit.

8) What role does the second section play in the full manuscript? Is it useful?

9) The results in the abstract and conclusions should be presented more simplified and qualitatively.

10) The test contents are not more and should enhance the analysis of test results.

6. PLOS authors have the option to publish the peer review history of their article (what does this mean?). If published, this will include your full peer review and any attached files.

Reviewer #1: No

Reviewer #2: No

---

## [Author Response · Author response to Decision Letter 0]

3 Oct 2022

Response to Reviewers’ Comments:

We are really thankful to you for your marvellous efforts in our manuscript and give us constructive and meaningful comments, which can overwhelmingly improve the quality of our paper. Therefore, we have carefully revised the manuscript word by word. Thanks again for your work on our manuscript. The revised contents are marked in red fonts in the present manuscript.

Journal Requirements:

2. We suggest you thoroughly copyedit your manuscript for language usage, spelling, and grammar. 

3. Please ensure that you have an ORCID iD and that it is validated in Editorial Manager. 

Response: Thank you for your advice. The file has been modified according to the PLOS ONE style template and named as required. We have thoroughly checked and revised the manuscript's language usage, spelling and grammar. The editing manager has verified my ORCID iD. I have made corrections and supplements as required.

The name of the colleague editing the manuscript: Jianfeng Yang, a university professor.

Reviewer Comments:

Reviewer 1

1. This article is not written in standard English, and the language needs to be improved greatly.

Response: Thank you for your advice. University professors have been invited to check and revise the overall English grammar of the thesis.

2. The first paragraph in Introduction seems to be irrelevant to this study.

Response: Thank you for your advice. I have rewritten the Introduction. 

3. What does "three under" coal mining mean in the first paragraph?

Response: Thank you for your advice. Relevant expressions have been deleted from the Introduction. "Three underground" coal mining refers to a general term for coal mining methods under buildings, railways and water bodies.

4. It's very confusing to talk about slurry filling and gangue filling in the article. Please clarify the relationship and what is the main research object.

Response: Thank you for your advice. The paper mainly refers to slurry filling and gangue slurry filling. The slurry filled with slurry is mainly cement, while the gangue slurry is mainly coal gangue ground into powder, that is, coal gangue. The main difference between the two is that the materials are different, and the corresponding physical and chemical properties are also different, which is described in detail in Section 4.1. Due to different filling purposes, gangue slurry filling differs from paste filling, and the filling strength is not too high. The amount of gangue filling is mainly increased by optimizing the slurry ratio and process. At the same time, to reduce the filling cost, the addition of cement-based materials is cancelled. However, the reduction of the content of fine aggregate leads to a great difference in the transport performance and flow performance between the coal gangue slurry and the traditional slurry.

Finally, the full text of the paper has been checked, and some inaccuracies have been corrected.

5. Line 57: “At present, scholars at home and abroad have put forward several grouting diffusion theories through assumptions and simplified methods, including penetration grouting, splitting grouting and compaction grouting, of which penetration grouting is the most commonly used.” Is there any literature basis for this statement, why is it more widely used, and whether it is more in line with engineering practice? Please explain the reason.

Response: Thank you for your advice. Relevant expressions have been deleted from the Introduction. But I can explain this problem. Currently, the seepage grouting method is widely used because of its low grouting pressure and small disturbance to the soil structure, especially in projects with high requirements for ground settlement control. For the engineering background described in the paper, the seepage grouting is more in line with the engineering practice.

6. Starting from line 63, references frequently use “It...” This is not a standard writing format, and the overall description of the paragraph is illogical. please revise carefully.

Response: Thank you for your advice. I have rewritten the Introduction. 

7. Introduction was written in a confused way. It did not describe and summarize the current situation of grouting research. It was impossible to see what was better than the existing research or what was considered more or what problems were solved.

Response: Thank you for your advice. I have rewritten the introduction.

Large scale and high-intensity mining activities and the utilization of coal resources have produced a large amount of solid waste of coal gangue [1,2], occupying a large number of land resources and polluting air, groundwater and soil, resulting in a series of ecological, environmental problems [4-6]. The coal gangue and water are mixed into a certain concentration of gangue slurry through the underground slurry preparation system, and the goaf caving zone is filled with adjacent grouting to realize the harmless disposal of coal gangue under the low interference condition of the working face[7,8]. The filtration effect mainly occurs in the flow process of suspension in porous media. When the slurry is injected into the rock and soil mass, some cement particles will be blocked by the particle skeleton, which will gradually filter out the cement particles, leading to the decrease of grout concentration, the blockage of pores, and the increase of grouting difficulty, which is called filtering effect [9-11]. The filtration effect generally exists in the seepage grouting of granular rock and soil, which plays an important role in seepage grouting [12-14]. The filtration effect significantly affects the gangue slurry velocity and concentration, making it difficult to evaluate the gangue slurry diffusion range.

The filtration effect in the slurry infiltration process is an essential factor affecting slurry diffusion [15-17]. The main research methods include four [18]: a macro model based on macro experimental phenomena, the study of the analysis method of single particle trajectory of suspension, the method of studying congestion through probability, and the mesh model method. The research on the filtering effect in soil mass infiltration grouting is developed based on the research of other disciplines. The filtering effect has been confirmed in the uniaxial sand soil infiltration grouting test [19,20]: the test shows that, as the filtered cement particles gradually block the soil mass pores, the grouting pressure gradually increases. At the same time, the concentration of grout injected into the soil mass is uneven, and the closer to the grouting mouth, the greater the concentration of grout. These test results show the importance of considering the filtration effect. Currently, there is relatively little research on the filtration effect in seepage grouting. Some scholars have conducted indoor tests to study the seepage effect's influence on grouting diffusion [21-23]. In terms of model, F Bouchelaghem [24] studied the micro mechanism of filtration effect、the coupling calculation model of fluid convection, hydrodynamic dispersion and other factors. However, because the mechanism of the seepage effect is very complex, it is impossible to obtain the change rule of a physical quantity in the process of mud diffusion comprehensively and intuitively through experimental methods. In terms of numerical calculation, some scholars have established iterative numerical calculation methods to consider the seepage effect from the perspective of mass conservation [25]. Or use the grid model method to study the sand column grouting model considering the seepage effect [26]. J. S. Kim et al. [27] gave an iterative numerical calculation method combining mass conservation equation and filtering law. S Maghous et al. [28] extended it to column infiltration grouting. The above research methods all consider the filtration effect from the perspective of quality conversion and have their advantages. However, most existing models are one-dimensional models based on indoor one-way grouting tests, which cannot be directly applied to on-site grouting. There is a lack of research on the fluidity characteristics of coal gangue slurry under the effect of seepage in goaf. Therefore, it is necessary to carry out an experimental study on seepage grouting of coal gangue slurry in the pore medium of goaf.

Therefore, based on the Darcy seepage law, a seepage theoretical calculation model considering the time-space effect of seepage is established, the "water-cement ratio change matrix" in the seepage process of coal gangue slurry is deduced, and the slurry particle deposition law is analyzed. The self-developed visual testing platform for mud diffusion in goaf is used for testing. The AHFO is used to monitor the flow and diffusion of coal gangue slurry in the collapse area of the goaf, and the gravity gradient and water cement ratio of the slurry in the goaf are measured. Finally, carry out an engineering test.

8. Line 129: “which has been identified as the main factor in controlling slurry infiltration grouting in many studies” Please list relevant research literature.

Response: Thank you for your advice. The corresponding research literature has been added in the corresponding position.

1.Zhu G, Zhang Q, Liu R, Bai J, Li W, Feng X. Experimental and Numerical Study on the Permeation Grouting Diffusion Mechanism considering Filtration Effects. GEOFLUIDS. 2021;2021. doi:10.1155/2021/6613990

2.Pan D, Hong K, Fu H, Zhou J, Zhang N, Lu G. Influence characteristics and mechanism of fragmental size of broken coal mass on the injection regularity of silica sol grouting. CONSTRUCTION AND BUILDING MATERIALS. 2021;269. doi:10.1016/j.conbuildmat.2020.121251

3.Liu J, Zhang X, Li M, Lan X, Hao P. Research on Permeation Grouting Mechanism Considering Gravity in the Treatment of Mud Inrush Disaster. POLISH JOURNAL OF ENVIRONMENTAL STUDIES. 2021;30: 751–762. doi:10.15244/pjoes/123924

9. Please analyze in detail what each item in formula (7) expresses and how to get the subsequent conclusion from this formula.

Response: Thank you for your advice. Relevant contents have been supplemented in Section 2.2.

 (7)

Where i and j are the unit length of grouting seepage distance and the unit length of the injected medium, i≥1, j≥1. It can be seen from the above formula that r11 is the initial water-cement ratio of the slurry, and ri1 ≤ r11, rij ≤ ri (j+1) ≤ ri (j+2) ≤... ≤ ri (j+m) ≤..., m ≥ 1. The water cement ratio of the grout increases along the seepage direction after the grout passes through the percolation. The farther away from the grouting pipe, the greater the water-cement ratio of the grout. It is especially pointed out that the water-cement ratio near the grouting mouth is smaller than the initial water-cement ratio.

10. How to verify that the established Goaf coal gangue slurry diffusion test platform is similar to the actual goaf situation, and how to prove that it can represent the actual grouting situation?

Response: Thank you for your advice. After the mining of the working face, the overlying strata in the goaf behind the working face will collapse, and the primary weighting of the basic roof will form an "O-X" fracture. After the basic roof cycle is broken, the rock blocks will form a masonry beam structure along the direction of the working face. The broken ends of the working face will form an arc triangle block, and a rock-bearing structure will be formed in the goaf. This structure can bear most of the load of the overlying strata. The collapse gangue at the lower part of the arc triangle block is in a loose accumulation state, and the gap between the rock blocks is large, which is a space that the slurry can fill. But outside the arc triangle block, that is, in the middle of the goaf in the dip direction, the rock block in the caving zone bears most of the load of the overburden of the stope, leading to the fact that the gangue in the caving zone in the middle of the goaf is compacted, the gap between the rock blocks is also small, and it is difficult to inject slurry, which makes it difficult for the slurry to fill and use the space.

For this test, the main simulation is the lower collapse zone of the arc triangle area in the goaf behind the working face, mainly the gangue in the uncompacted area, which is generally only affected by its weight. It is difficult to accurately simulate the gangue in the goaf of the coal mine, which can only be qualitatively simulated.

Actual goaf conditions. Although there are some differences between model test and actual engineering conditions, a series of previous studies have proved that model test is an effective method to study grouting problems.

Neighbourhood grouting and filling refers to the arrangement of adjacent grouting and filling boreholes in the upper area of the collapse zone to build a slurry filling channel in the inclined space in the same layer. The main means of implementation is to construct the upward drilling from the adjacent working face roadway or adjacent main roadway to the upper area of the collapse zone, as shown in Fig 1. The application conditions of adjacent grouting and filling are relatively broad, which can be applied to the working face under mining to achieve filling while mining and can also be used to achieve post-mining filling by constructing an overhead borehole in the main roadway or stoping line as a technical means of slurry filling in the old goaf. As shown in Fig2 in the paper, holes are opened on the right side of the test box to simulate the actual situation on site, and the grouting pipe is inclined at a certain angle for grouting.

Fig 1. Schematic diagram of adjacent grouting filling technology

To prove the reliability of the test and test results, the industrial field test is supplemented.

5 Application of adjacent grouting filling of gangue slurry

5.1 Project overview

The raw coal production capacity of the well is 10 Mt/a, and the total amount of gangue is about 0.85 Mt/a producing the pure gangue produced bottom raising and chamber excavation during the layout of the underground working face is about 20 Mt/a.

5.2 Filling Scheme

According to the preliminary test results, it is determined that the mass concentration of gangue slurry in the project application stage is 70%, the pumping pressure is 8 Mpa, the working flow rate is 1.5 m/s, the selected plunger filling pump model is HBMG30/21-110S, and the selected mixer model is JSY3200. The process is divided into the following steps: quantitative feeding → mixing and pulping → pumping → filling.

Firstly, the crushing, pulping and filling equipment is arranged in the machine head chamber of working face 301 by using the abandoned roadway of the mine. The underground raw gangue is crushed in two stages to form a powder with a particle size of less than 3 mm. Then it is made into 70% slurry by adding the existing underground mine water through a quantitative feeder. Then it is transported to the adjacent borehole with a plunger pump and pipeline for filling, as shown in Fig 11.

Based on the theoretical calculation, measured data, and the analysis of the roof condition of the working face, the height of the gob collapse zone formed after the coal seam mining in working face 301 is 9.2~16.9 m. Therefore, to ensure that the final hole is located in the collapse zone, the final hole height of the filling borehole is 11m, the elevation angle is 9 °, and the borehole diameter is designed in the early stage 133 mm. The protective casing is placed in the whole process of grouting drilling to isolate the gangue from the gob. The end of the grouting pipe is 3.0~5.0 m connected with the flower pipe to increase the grouting range, and the hole diameter of the flower hole is 25 mm.

(a) Pulping station (b) Filling borehole

Fig 11. Site of adjacent grouting filling.

5.3 Filling effect

The on-site pressure monitoring data shows that the pressure of the pipeline and filling borehole is stable during the grouting period, which can realize continuous and stable filling operation, and the grouting and filling effect adjacent to the goaf is good. The feasibility of the underground adjacent grouting filling technology is successfully verified, which further proves that there is sufficient filling space in the gob collapse zone and that the equipment and transportation parameters selected for engineering application are reasonable, and the system stability is high.

During the filling period, the transportation state of the gangue slurry pipeline is stable. The solid particles in the slurry have not silted up, the slurry has good fluidity in the void space of the goaf, and there is no aggregation and settlement phenomenon, indicating the transportation flow rate and slurry grading concentration of the slurry pipeline designed in the early stage is reasonable.

When the cumulative grouting volume of a single hole is 700 m3, the grout diffusion distance is expected to be 51 m according to the gravity flow gradient of the grout in the goaf of 6.34%. Pass through the inclined hole of coal pillar construction at the position 45 m away from the borehole, and the hole depth is 36m. Ensure that the final hole of the borehole is located above the bottom plate. Immediately after the hole is formed, use the borehole peep to peep at the situation in the hole. The peeping situation in the hole is shown in Fig 12. The results of borehole peeping show that the lens of the peeping mirror is blocked by the slurry, indicating that the coal gangue slurry has spread to this position. The filling slurry has good flow and diffusion performance in the goaf, and the flow law conforms to the test results. The slurry diffusion distance is greater than 45 m.

(a) Gangue free slurry (b) Gangue slurry

Fig 12. Bored peeking pictures.

11. In Section 4.4, how many sampling points are there in the experiment? Please show the location of the sampling points with a diagram. Can the number meet the requirements of Kriging interpolation?

Response: Thank you for your advice. As shown in Fig. 2, according to the layout scheme of optical fiber in the test model and the sampling interval of the DTS demodulator of 0.1m, one sampling point shall be taken every 10 cm along the optical fiber. There are 40 sampling points in the whole model test. 

Fig 2. Arrangement method of active heating optical fiber.

On this basis, this paper optimizes the number of sampling points by using the SAS program provided by skopp et al. [1] and selects seven sampling points on the premise of balancing accuracy and calculation speed. 

The optimal number of samples to be collected depends on the variability of the total number of samples, the accuracy level required to estimate the mean value of the total number of samples, the confidence interval required to estimate the mean value of the total number of samples, the cost of sampling, the consideration of sample analysis, and the available labor and equipment. The estimation of each point does not need to use all sample values because using too much data will increase the amount of calculation, but the accuracy is not significantly improved. At the same time, using long-distance data points may be more likely to violate the assumption of second-order stability. When the samples conform to the normal distribution and are independent of each other, the following formula is commonly used to estimate the required number of samples to achieve the required accuracy of the estimated mean value of the studied variables:

 (1)

Where t is a confidence interval on both sides under a given probability, and the t value can be found from the statistical table; s is an initial estimate of the standard variance of the total sample, and d is the allowable deviation between the mean value of the total sample and the mean value of the measured value. 

[1] Skopp J, Kachman S D, Hergert G W. Comparison of procedures for estimating sample numbers[J]. Communications in soil science and plant analysis, 1995, 26(15-16): 2559-2568.

12. The grouting pressure is an important factor, but it is not described in Section 4.3.

Response: Thank you for your advice. It has been supplemented in Section 3 of the paper to introduce the specific value of grouting pressure. 

The grouting pressure of this test is 0.032 MPa.

13. The display of flow process of slurry in Figure 8 is too little to show the specific situation.

Response: Thank you for your advice. The graph of slurry flow process has been added.

(a)Vertical expansion (b)Lateral diffusion

(c)Gangue slurry diffusion (d)Filling completed

Fig 8. Flow process of gangue slurry in gangue accumulation.

14. What do the various colors in Figure 10 mean?

Response: Thank you for your advice. The colour change in Fig 10 represents the size of the temperature value, and Fig 10 has been modified. 

(a) (b)

(c) (d)

Fig 10. Grouting diffusion range.

Reviewer 2

The submission investigates the grouting mechanism and experimental study of goaf. It is an interesting work which is a hot research topic recently. In general, this work was well prepared but the English language need to be polished by a native English speaker. Although the paper presents experimental results that are potentially useful, the study has several limitations. The limitations, in my opinion, are as follows:

1. The abstract should be polished and stress the main points.

Response: Thank you for your advice. I have rewritten the abstract.

The filtration effect significantly affects the gangue slurry velocity and concentration, making it difficult to evaluate the gangue slurry diffusion range. Based on the Darcy seepage law, a seepage theoretical calculation model is established considering the filtration time and space effect. And the "water-cement ratio change matrix" in the seepage process of coal gangue slurry is deduced, revealing the basic mechanism of the porous media filtration effect, and the water-cement ratio gradually increases in the seepage process of gangue slurry. The visual test platform for slurry diffusion in goaf was independently developed for testing. The active heating optical fiber method (AHFO) was used to monitor the flow and diffusion of coal gangue slurry in the collapse zone of goaf, and the gravity gradient and water cement ratio of slurry in goaf were measured. The law of particle sedimentation in the gangue slurry flow process under the filtration effect was revealed, and engineering verification was carried out. The results show that the average slope of the gangue slurry in the gangue accumulation is 6.34%, and the overall flow law of the gangue slurry in the goaf is the first longitudinal expansion and then transverse diffusion. The water-cement ratio near the grouting mouth is smaller than the initial water-cement ratio, the near-end water-cement ratio is smaller, and the far-end water-cement ratio is larger. During on-site filling, the accumulated grouting volume of a single hole is 700 m3, and the gangue slurry diffusion distance is greater than 45m, indicating that the gangue slurry has good fluidity.

2. The presentation level is not adequate, especially the Introduction. The outcomes of the past researches and their relation to the present study should be clearly presented. At present, the Introduction seems like a string of studies that were done on the topic rather than a meaningful narrative.

Response: Thank you for your advice. I have rewritten the introduction.

Large scale and high-intensity mining activities and the utilization of coal resources have produced a large amount of solid waste of coal gangue [1,2], occupying a large number of land resources and polluting air, groundwater and soil, resulting in a series of ecological, environmental problems [4-6]. The coal gangue and water are mixed into a certain concentration of gangue slurry through the underground slurry preparation system, and the goaf caving zone is filled with adjacent grouting to realize the harmless disposal of coal gangue under the low interference condition of the working face[7,8]. The filtration effect mainly occurs in the flow process of suspension in porous media. When the slurry is injected into the rock and soil mass, some cement particles will be blocked by the particle skeleton, which will gradually filter out the cement particles, leading to the decrease of grout concentration, the blockage of pores, and the increase of grouting difficulty, which is called filtering effect [9-11]. The filtration effect generally exists in the seepage grouting of granular rock and soil, which plays an important role in seepage grouting [12-14]. The filtration effect significantly affects the gangue slurry velocity and concentration, making it difficult to evaluate the gangue slurry diffusion range.

The filtration effect in the slurry infiltration process is an essential factor affecting slurry diffusion [15-17]. The main research methods include four [18]: a macro model based on macro experimental phenomena, the study of the analysis method of single particle trajectory of suspension, the method of studying congestion through probability, and the mesh model method. The research on the filtering effect in soil mass infiltration grouting is developed based on the research of other disciplines. The filtering effect has been confirmed in the uniaxial sand soil infiltration grouting test [19,20]: the test shows that, as the filtered cement particles gradually block the soil mass pores, the grouting pressure gradually increases. At the same time, the concentration of grout injected into the soil mass is uneven, and the closer to the grouting mouth, the greater the concentration of grout. These test results show the importance of considering the filtration effect. Currently, there is relatively little research on the filtration effect in seepage grouting. Some scholars have conducted indoor tests to study the seepage effect's influence on grouting diffusion [21-23]. In terms of model, F Bouchelaghem [24] studied the micro mechanism of filtration effect、the coupling calculation model of fluid convection, hydrodynamic dispersion and other factors. However, because the mechanism of the seepage effect is very complex, it is impossible to obtain the change rule of a physical quantity in the process of mud diffusion comprehensively and intuitively through experimental methods. In terms of numerical calculation, some scholars have established iterative numerical calculation methods to consider the seepage effect from the perspective of mass conservation [25]. Or use the grid model method to study the sand column grouting model considering the seepage effect [26]. J. S. Kim et al. [27] gave an iterative numerical calculation method combining mass conservation equation and filtering law. S Maghous et al. [28] extended it to column infiltration grouting. The above research methods all consider the filtration effect from the perspective of quality conversion and have their advantages. However, most existing models are one-dimensional models based on indoor one-way grouting tests, which cannot be directly applied to on-site grouting. There is a lack of research on the fluidity characteristics of coal gangue slurry under the effect of seepage in goaf. Therefore, it is necessary to carry out an experimental study on seepage grouting of coal gangue slurry in the pore medium of goaf.

Therefore, based on the Darcy seepage law, a seepage theoretical calculation model considering the time-space effect of seepage is established, the "water-cement ratio change matrix" in the seepage process of coal gangue slurry is deduced, and the slurry particle deposition law is analyzed. The self-developed visual testing platform for mud diffusion in goaf is used for testing. The AHFO is used to monitor the flow and diffusion of coal gangue slurry in the collapse area of the goaf, and the gravity gradient and water cement ratio of the slurry in the goaf are measured. Finally, carry out an engineering test.

3. In line 32, AHFO is not suitable to be one key word.

Response: Thank you for your advice. It has been modified to Actively Heated Fiber Optic.

4. It is better to reconsider the title of each section.

Response: Thank you for your advice. 

2.1 Gangue slurry filling technology in goaf

2.2 Analysis of influence mechanism of space-time effect of grouting filtration

4 Flow performance of gangue slurry

4.2 Filtration time-space effect of gangue slurry flow process

5 How to reflect the field performance by the results of test platform? Is there any difference?

Response: Thank you for your advice. According to the test, an important parameter is obtained: the slurry's average slope is 6.34%. This parameter is used to estimate the grouting diffusion range on site, and the accuracy of this parameter is verified by drilling peeping technology. Of course, the simulation experiment has obtained some regular results, which are limited by technical means and have not been verified in the field. For the accuracy of the simulation experiment, the most important thing is to consider the size effect on the results. However, it has not been well resolved.

5 Application of adjacent grouting filling of gangue slurry

5.1 Project overview

The raw coal production capacity of the well is 10 Mt/a, and the total amount of gangue is about 0.85 Mt/a producing the pure gangue produced bottom raising and chamber excavation during the layout of the underground working face is about 20 Mt/a.

5.2 Filling Scheme

According to the preliminary test results, it is determined that the mass concentration of gangue slurry in the project application stage is 70%, the pumping pressure is 8 Mpa, the working flow rate is 1.5 m/s, the selected plunger filling pump model is HBMG30/21-110S, and the selected mixer model is JSY3200. The process is divided into the following steps: quantitative feeding → mixing and pulping → pumping → filling.

Firstly, the crushing, pulping and filling equipment is arranged in the machine head chamber of working face 301 by using the abandoned roadway of the mine. The underground raw gangue is crushed in two stages to form a powder with a particle size of less than 3 mm. Then it is made into 70% slurry by adding the existing underground mine water through a quantitative feeder. Then it is transported to the adjacent borehole with a plunger pump and pipeline for filling, as shown in Fig 11.

Based on the theoretical calculation, measured data, and the analysis of the roof condition of the working face, the height of the gob collapse zone formed after the coal seam mining in working face 301 is 9.2~16.9 m. Therefore, to ensure that the final hole is located in the collapse zone, the final hole height of the filling borehole is 11m, the elevation angle is 9 °, and the borehole diameter is designed in the early stage 133 mm. The protective casing is placed in the whole process of grouting drilling to isolate the gangue from the gob. The end of the grouting pipe is 3.0~5.0 m connected with the flower pipe to increase the grouting range, and the hole diameter of the flower hole is 25 mm.

(a) Pulping station (b) Filling borehole

Fig 11. Site of adjacent grouting filling.

5.3 Filling effect

The on-site pressure monitoring data shows that the pressure of the pipeline and filling borehole is stable during the grouting period, which can realize continuous and stable filling operation, and the grouting and filling effect adjacent to the goaf is good. The feasibility of the underground adjacent grouting filling technology is successfully verified, which further proves that there is sufficient filling space in the gob collapse zone and that the equipment and transportation parameters selected for engineering application are reasonable, and the system stability is high.

During the filling period, the transportation state of the gangue slurry pipeline is stable. The solid particles in the slurry have not silted up, the slurry has good fluidity in the void space of the goaf, and there is no aggregation and settlement phenomenon, indicating the transportation flow rate and slurry grading concentration of the slurry pipeline designed in the early stage is reasonable.

When the cumulative grouting volume of a single hole is 700 m3, the grout diffusion distance is expected to be 51 m according to the gravity flow gradient of the grout in the goaf of 6.34%. Pass through the inclined hole of coal pillar construction at the position 45 m away from the borehole, and the hole depth is 36m. Ensure that the final hole of the borehole is located above the bottom plate. Immediately after the hole is formed, use the borehole peep to peep at the situation in the hole. The peeping situation in the hole is shown in Fig 12. The results of borehole peeping show that the lens of the peeping mirror is blocked by the slurry, indicating that the coal gangue slurry has spread to this position. The filling slurry has good flow and diffusion performance in the goaf, and the flow law conforms to the test results. The slurry diffusion distance is greater than 45 m.

(a) Gangue free slurry (b) Gangue slurry

Fig 12. Bored peeking pictures.

6 Currently, the Conclusion presents a relatively verbose summary of the results – I recommend making the results more general and especially pointing out the practical implications of the research.

Response: Thank you for your advice. The conclusion has been revised. According to the calculation of the gravity gradient of the slurry in the goaf of 6.73%, the slurry diffusion distance is expected to be 51 m. At the position 45 m away from the drill hole, the dip hole was constructed through the coal pillar. The peeping results showed that the hole wall at the front end of the drill hole was complete, the peep meter continued to go deep to the end of the drill hole, and the lens of the peep meter was covered with slurry, indicating that the gangue slurry had spread to the peeping position. It is proved that the filling slurry has good flow and diffusion performance in the goaf, and the diffusion distance of the filling drilling slurry is greater than 45 m.

Conclusion

(1) Through theoretical analysis, the "water-cement ratio change matrix" in the coal gangue slurry seepage process is deduced to reveal the basic mechanism of the porous media filtration effect, and the water-cement ratio gradually increases in the seepage process of gangue slurry. 

(2) The overall flow law of the gangue slurry in the goaf shows the law of first longitudinal expansion and then horizontal diffusion. After stabilization, the average slope of the gangue slurry in the gangue stack is 6.34%.

(3) Filtration effect has an obvious space-time effect. The proportion of gangue particles less than 200 mesh in the gangue slurry near the grouting mouth is 9.81%, while the far end position is 20.36%. In the direction parallel to the infiltration direction, the water-cement ratio gradually increases with the diffusion distance.

(4) The engineering application results show that the cumulative grouting volume of a single hole is 700 m3, the slurry diffusion distance is greater than 45 m, and the slurry has excellent fluidity in the void space of the goaf.

7 There have numerous format problems, eg. always missing a gap between a number and unit.

Response: Thank you for your advice. The papers have been checked and revised all day according to the format requirements of the journal. 

8 What role does the second section play in the full manuscript? Is it useful?

Response: Thank you for your advice. Section 2.1 introduces the technical process of coal gangue slurry filling and the adjacent grouting method. The subsequent test and project site are based on this technical process and method. Section 2.2 mainly derives the seepage theoretical calculation model considering seepage's time and space effect based on Darcy's seepage law and the equivalent relationship between seepage and pore volume. The "water-cement ratio change matrix" in the process of mud seepage is derived. The basic mechanism of the porous media seepage effect is analyzed. Theoretical analysis shows that the seepage effect has an obvious space-time effect related to seepage time and diffusion distance. The water-cement ratio near the grouting mouth is smaller than the initial water-cement ratio, the near-end water-cement ratio is smaller, and the far-end water-cement ratio is larger. The following test results verify the correctness of this conclusion. 

 (7)

Where i and j are the unit length of grouting seepage distance and the unit length of the injected medium, i≥1, j≥1. It can be seen from the above formula that r11 is the initial water-cement ratio of the slurry, and ri1 ≤ r11, rij ≤ ri (j+1) ≤ ri (j+2) ≤... ≤ ri (j+m) ≤..., m ≥ 1. The water cement ratio of the grout increases along the seepage direction after the grout passes through the percolation. The farther away from the grouting pipe, the greater the water-cement ratio of the grout. It is especially pointed out that the water-cement ratio near the grouting mouth is smaller than the initial water-cement ratio.

Sections 2.1 and 2.2 have been supplemented and modified.

9 The results in the abstract and conclusions should be presented more simplified and qualitatively.

Response: Thank you for your advice. The summary and conclusions have been refined. 

Abstract

The filtration effect significantly affects the gangue slurry velocity and concentration, making it difficult to evaluate the gangue slurry diffusion range. Based on the Darcy seepage law, a seepage theoretical calculation model is established considering the filtration time and space effect. And the "water-cement ratio change matrix" in the seepage process of coal gangue slurry is deduced, revealing the basic mechanism of the porous media filtration effect, and the water-cement ratio gradually increases in the seepage process of gangue slurry. The visual test platform for slurry diffusion in goaf was independently developed for testing. The active heating optical fiber method (AHFO) was used to monitor the flow and diffusion of coal gangue slurry in the collapse zone of goaf, and the gravity gradient and water cement ratio of slurry in goaf were measured. The law of particle sedimentation in the gangue slurry flow process under the filtration effect was revealed, and engineering verification was carried out. The results show that the average slope of the gangue slurry in the gangue accumulation is 6.34%, and the overall flow law of the gangue slurry in the goaf is the first longitudinal expansion and then transverse diffusion. The water-cement ratio near the grouting mouth is smaller than the initial water-cement ratio, the near-end water-cement ratio is smaller, and the far-end water-cement ratio is larger. During on-site filling, the accumulated grouting volume of a single hole is 700 m3, and the gangue slurry diffusion distance is greater than 45m, indicating that the gangue slurry has good fluidity.

Conclusion

(1) Through theoretical analysis, the "water-cement ratio change matrix" in the coal gangue slurry seepage process is deduced to reveal the basic mechanism of the porous media filtration effect, and the water-cement ratio gradually increases in the seepage process of gangue slurry. 

(2) The overall flow law of the gangue slurry in the goaf shows the law of first longitudinal expansion and then horizontal diffusion. After stabilization, the average slope of the gangue slurry in the gangue stack is 6.34%.

(3) Filtration effect has an obvious space-time effect. The proportion of gangue particles less than 200 mesh in the gangue slurry near the grouting mouth is 9.81%, while the far end position is 20.36%. In the direction parallel to the infiltration direction, the water-cement ratio gradually increases with the diffusion distance.

(4) The engineering application results show that the cumulative grouting volume of a single hole is 700 m3, the slurry diffusion distance is greater than 45 m, and the slurry has excellent fluidity in the void space of the goaf.

10 The test contents are not more and should enhance the analysis of test results.

Response: Thank you for your advice. Section 4.2 and 4.3 of the paper supplement the in-depth analysis of the test results.

The main reasons for this phenomenon are: the slurry is mainly distributed in the cavities composed of each rock block and the voids between each rock block. The slurry in different cavities is in a polarized state. When the cavity is filled with slurry, the voids between rock blocks around the cavity are filled with slurry, but there are still some small voids without a slurry. When there is no slurry in the cavity, the slurry in the interstices between the rock blocks around the cavity is distributed as a broken branch. That is, the slurry blocks the crevices between the rock blocks (there are large particles in the slurry or the crevices between the rock blocks are too small), indicating that the slurry flow in the interstices between the rock blocks is selective. 

(a)Vertical expansion (b)Lateral diffusion

(c)Gangue slurry diffusion (d)Filling completed

Fig 8. Flow process of gangue slurry in gangue accumulation.

I am very grateful to the editorial department and the reviewers for their advice, which allowed me to carefully revise the manuscript, reorganize the paper, modify the keywords, title, and content, and take this opportunity to improve the academic quality and readability of the manuscript. Thanks again to the editorial department and the reviewers for their work. Your comments are all valuable and very helpful for revising and improving our paper.

Dr. GU

Xi'an City, Shaanxi Province, China 

October 1, 2022

---

## [Decision Letter · Decision Letter 1]

23 Jan 2023

PONE-D-22-23513R1Grouting mechanism and experimental study of goaf considering filtration effectPLOS ONE

Dear Dr. GU,

Thank you for submitting your manuscript to PLOS ONE. After careful consideration, we feel that it has merit but does not fully meet PLOS ONE’s publication criteria as it currently stands. Therefore, we invite you to submit a revised version of the manuscript that addresses the points raised during the review process.

ACADEMIC EDITOR:<ul> <li> 

I carefully read through your manuscript. I feel that its content is enough to be acceptable for publication, but its writing-up should be significantly improved before any acceptance for publication. My comments are as follows:

1, Organization: Your focus is to determine the range of injected gangue slurry through its fluidity analysis. The content includes theoretical analysis and indoor experiment. What is their linkage between the two parts? The theoretical results should be compared with your experimental results.

2, theoretical analysis: This part is really confusing for reading because the formulae are not presented in a logical way. The principles or citations for the formulae development should be given.

3, application to a project: Is this application example used to check your experimental observation or your theory? It should be analyzed or explained to serve your focus of this manuscript.

4, English and presentations: Manuscript has many unclear statements or presentations. The novelty is not clearly presented. The citations in t5he text and references should follow the Journal style. Full revision or check should be done.

5, Conclusions: The items should be supported by data in your manuscript and enhanced to general knowledge. They will be useful for engineering practice or the understanding of injection flow mechanisms.

We look forward to receiving your revised manuscript.

Kind regards,

Jianguo Wang, PhD

Academic Editor

PLOS ONE

Journal Requirements:

Reviewers' comments:

Reviewer's Responses to Questions

**Comments to the Author**

1. If the authors have adequately addressed your comments raised in a previous round of review and you feel that this manuscript is now acceptable for publication, you may indicate that here to bypass the “Comments to the Author” section, enter your conflict of interest statement in the “Confidential to Editor” section, and submit your "Accept" recommendation.

Reviewer #1: All comments have been addressed

Reviewer #2: All comments have been addressed

2. Is the manuscript technically sound, and do the data support the conclusions?

Reviewer #1: Yes

Reviewer #2: Yes

3. Has the statistical analysis been performed appropriately and rigorously? 

Reviewer #1: Yes

Reviewer #2: Yes

4. Have the authors made all data underlying the findings in their manuscript fully available?

Reviewer #1: Yes

Reviewer #2: Yes

5. Is the manuscript presented in an intelligible fashion and written in standard English?

Reviewer #1: Yes

Reviewer #2: Yes

6. Review Comments to the Author

Reviewer #1: The author has made good modifications to the manuscript and all comments have been addressed, which conforms to the published standard

Reviewer #2: I am satisfied with the responses provided by the authors. The edits make it much easier to read the paper. I recommend the paper for acceptance.

7. PLOS authors have the option to publish the peer review history of their article (what does this mean?). If published, this will include your full peer review and any attached files.

Reviewer #1: No

Reviewer #2: No

---

## [Author Response · Author response to Decision Letter 1]

8 Feb 2023

Response to Reviewers’ Comments:

We are really thankful to you for your marvellous efforts in our manuscript and give us constructive and meaningful comments, which can overwhelmingly improve the quality of our paper. Therefore, we have carefully revised the manuscript word by word. Thanks again for your work on our manuscript. The revised contents are marked in red fonts in the present manuscript.

ACADEMIC EDITOR:

I carefully read through your manuscript. I feel that its content is enough to be acceptable for publication, but its writing-up should be significantly improved before any acceptance for publication. My comments are as follows:

1 Organization: Your focus is to determine the range of injected gangue slurry through its fluidity analysis. The content includes theoretical analysis and indoor experiment. What is their linkage between the two parts? The theoretical results should be compared with your experimental results.

Response: Thank you for your advice. The theoretical analysis qualitatively shows that the water-cement ratio of slurry increases with the increase of diffusion distance under the effect of infiltration. In the laboratory test, by measuring the mass fraction of gangue particles at different locations after grouting, it was found that the mass fraction of coarse gangue particles at the location far from the grouting pipe was significantly reduced. This means that the water-cement ratio of the slurry at the position far away from the grouting pipe increases, which verifies the correctness of the theoretical analysis. Relevant contents have been added in sections 2.2 and 4.2 of the paper for the supplementary explanation.

2 Theoretical analysis: This part is really confusing for reading because the formulae are not presented in a logical way. The principles or citations for the formulae development should be given.

Response: Thank you for your advice. The theoretical part of section 2.2 of the paper was reorganized and written, and some important formulas were added to ensure the logic and correctness of theoretical derivation.

3 Application to a project: Is this application example used to check your experimental observation or your theory? It should be analyzed or explained to serve your focus of this manuscript.

Response: Thank you for your advice. According to the permeability gradient obtained from the test and the height of the grouting pipe, the grouting diffusion distance is predicted to be 51m. When the cumulative grouting volume of a single hole is 700 m3, the grout diffusion distance is expected to be 51 m according to the gravity flow gradient of the grout in the goaf of 6.34%. The accuracy of slurry seepage slope in goaf obtained from laboratory test is verified. 

4 English and presentations: Manuscript has many unclear statements or presentations. The novelty is not clearly presented. The citations in the text and references should follow the Journal style. Full revision or check should be done.

Response: Thank you for your advice. The file has been modified according to the PLOS ONE style template and named as required. We have thoroughly checked and revised the manuscript's language usage, spelling and grammar. 

5 Conclusions: The items should be supported by data in your manuscript and enhanced to general knowledge. They will be useful for engineering practice or the understanding of injection flow mechanisms.

Response: Thank you for your advice. According to your suggestion, I have refined the conclusion again and revised it as follows:

(1) Through theoretical analysis, the "water-cement ratio change matrix" in the coal gangue slurry seepage process is deduced to reveal the basic mechanism of the porous media filtration effect, and the water-cement ratio gradually increases in the seepage process of gangue slurry. 

(2) The overall flow law of the gangue slurry in the goaf shows the law of first longitudinal expansion and then horizontal diffusion. Before the slurry is stabilized, the seepage gradient is large, and after stabilization, the seepage gradient decreases, the average seepage gradient is 6.34%.

(3) Filtration effect has an obvious space-time effect. The proportion of gangue particles less than 200 mesh in the gangue slurry near the grouting mouth is 9.81%, while the far end position is 20.36%. In the direction parallel to the infiltration direction, the water-cement ratio gradually increases with the diffusion distance. The farther away from the grouting mouth, the greater the water-cement ratio of the slurry.

(4) According to the permeability gradient obtained from the test and the height of the grouting pipe, the grouting diffusion distance is predicted to be 51m. The engineering application results show that the cumulative grouting volume of a single hole is 700 m3, the slurry diffusion distance is greater than 45 m, and the slurry has excellent fluidity in the void space of the goaf.

I am very grateful to the editorial department and the reviewers for their advice, which allowed me to carefully revise the manuscript, reorganize the paper, modify the keywords, title, and content, and take this opportunity to improve the academic quality and readability of the manuscript. Thanks again to the editorial department and the reviewers for their work. Your comments are all valuable and very helpful for revising and improving our paper.

Dr. GU

Xi'an City, Shaanxi Province, China 

February 8, 2023

---

## [Editor Report · Decision Letter 2]

10 Feb 2023

Grouting mechanism and experimental study of goaf considering filtration effect

PONE-D-22-23513R2

Dear Dr. GU,

We’re pleased to inform you that your manuscript has been judged scientifically suitable for publication and will be formally accepted for publication once it meets all outstanding technical requirements.

Kind regards,

Jianguo Wang, PhD

Academic Editor

PLOS ONE
---

## [Editor Report · Acceptance letter]

15 Feb 2023

PONE-D-22-23513R2 

Grouting mechanism and experimental study of goaf considering filtration effect 

Dear Dr. Gu:

I'm pleased to inform you that your manuscript has been deemed suitable for publication in PLOS ONE. Congratulations! Your manuscript is now with our production department. 

Kind regards, 

on behalf of

Dr. Jianguo Wang 

Academic Editor

PLOS ONE